# The Self-Supporting NiMn-LDHs/rGO/NF Composite Electrode Showing Much Enhanced Electrocatalytic Performance for Oxygen Evolution Reaction

Jia Wang and Yongfu Lian *

Key Laboratory of Functional Inorganic Material Chemistry, Ministry of Education, School of Chemistry and Materials Science, Heilongjiang University, Harbin 150080, China; 2201337@s.hlju.edu.cn
* Correspondence: chyflian@hlju.edu.cn

**Abstract:** The poor conductivity and instability of layered dihydroxides (LDHs) limit their widespread application in oxygen evolution reaction (OER). In this study, the composite electrode of NiMn-LDHs, reduced graphene oxide (rGO) and nickel foam (NF), i.e., NiMn-LDHs/rGO/NF, was prepared by a hydrothermal method. When subjected to oxygen evolution reaction (OER) catalytic performance in a solution of 1 M KOH, the NiMn-LDHs/rGO/NF composite catalyst exhibited an overpotential of only 140 mV at a current density of 10 mA cm$^{-2}$ and a Tafel slope of 49 mV dec$^{-1}$, which is not only better than the comparing RuO$_2$/NF catalyst, but also better than most of the Mn-based and the Ni–Fe-containing bimetallic OER catalysts reported in the literature. The excellent electrocatalytic performance is ascribed to the efficient integration of ultrathin NiMn-LDH sheets, thin-layered rGO and NF, contributing significantly to the decrease in charge transfer resistance and the increase in electrochemically active surface area. Moreover, NF plays a role of current collector and a role of rigid support for the NiMn-LDHs/rGO composite, contributing extra conductivity and stability to the NiMn-LDHs/rGO/NF composite electrode.

**Keywords:** self-supporting electrode; reduced graphene oxide; layered NiMn-dihydroxide; electrocatalyst; oxygen evolution reaction





## 1. Introduction

Hydrogen is an efficient and renewable energy resource, which is regarded to be a highly promising alternative to fossil fuel. It can help mitigate environmental pollution resulting from the combustion of fossil fuel, and meet the energy demand of our rapidly developing society. Among the many methods of hydrogen production, water electrolysis has attracted widespread attention as a high-efficiency and environmentally friendly technology to produce hydrogen. Electrolysis of water is composed of hydrogen evolution reaction (HER) in a two-electron process, and oxygen evolution reaction (OER) in a four-electron process [1]. Since the anodic four-electron reaction process of OER is much slower than that of the two-electron reaction process of HER, high overpotential and the sluggish kinetics inevitably occur in the electrolysis of water. Therefore, the development of an effective electrocatalyst to improve the energy conversion efficiency is needed [2,3]. However, the slow kinetics of the OER presents a significant challenge for the widespread implementation of water electrolysis as a large-scale hydrogen production technology. Therefore, it is crucial to develop an efficient OER catalyst.

LDHs are two-dimensional layered functional materials composed of positively charged hydroxide layers with equilibrium charges of anions between the layers [4]. The general chemical formula of LDHs is $[M_{1-x}^{2+}M_x^{3+}(OH)_2]^{x+}[A^{n-}]_{x/n} \cdot zH_2O$, in which z is the number of interlayer water molecules [5]. LDHs, as common catalysts, have a unique two-dimensional layered structure and could be exfoliated into ultrathin nanosheets, which

are characterized by their large specific surface area and are conducive to the improvement in the catalytic performance of LDHs. On the other hand, the metal cations in LDH materials have variable valence states, which is beneficial for the electron transfer in the catalytic reactions. To date, many LDHs, including NiFe-LDHs, FeCo-LDHs, NiCo-LDHs, NiCr-LDHs and NiMn-LDHs have been prepared and applied in the fields of catalysts, adsorption materials, energy storage materials and sensor materials [6–11], etc. Recently, NiMn-LDH was reported to be of quite good catalytic activity in the OER reaction when applied as catalyst in the electrolysis of water [12]. Wang et al. synthesized cobalt-doped NiMn-LDH by a hydrothermal method, which exhibited an overpotential of 310 mV and a Tafel slope of 59 mV dec$^{-1}$ at a current density of 10 mA cm$^{-2}$ in alkaline media [13]. Nonetheless, the electrocatalytic performance of LDH materials is still limited for their poor electrical conductivity and easy agglomeration leaded low surface area (ECSA) [14–17].

Because of its unique ultrathin sheet structure and excellent chemical stability and electrical conductivity, graphene is often integrated with some catalysts to improve their catalytic performance [18]. The composite of cobalt-based metal-organic framework (MOF) and reduced graphene oxide (rGO) synthesized by Yaqoob et al. displayed good electrochemical activity with an overpotential of 290 mV at 10 mA cm$^{-2}$ [19]. The composite of graphene and $Co_3O_4$ prepared by Mao et al. demonstrated an improved OER performance, owing to the heterojunction created at the interface of graphene sheet and $Co_3O_4$ nanoparticles [20]. The composite of copper benzodicar boxylate and rGO prepared by Jahan et al. was a promising composite catalyst for OER because of its quite good stability and much low resistance [21]. Han et al. prepared the composite of FeNi-LDH and GO by GO guiding the construction of FeNi-LDH arrays. When applied as a desirable bifunctional electrocatalyst for the splitting of water, it demonstrated a better OER performance than a conventional commercial catalyst because of its enhanced conductivity and electronic interactions [22]. Mooni et al. prepared the bimetal oxides ($MnO_2$-NiO) and graphene oxide-mixed composite electrodes (GO/$MnO_2$-NiO), which displayed an overpotential of 379 mV and maintained stable for 8 h at 10 mA cm$^{-2}$ when applied in the study of anodic water oxidation activities in an aqueous alkaline solution [23]. Ma et al. firstly synthesized NiMn-LDH nanoplatelets by a hydrothermal treatment of a mixed $Ni^{2+}$/$Mn^{2+}$ salt solution in the presence of $H_2O_2$ and hexamethylenetetramine, and then prepared the hetero-assembly of NiMn-LDH nanosheets and GO/rGO through a molecular hybridization of the LDH nanosheets with rGO. When applied as an electrocatalyst for OER, the face-to-face hetero-assembly of NiMn LDH nanosheets with conductive rGO at an alternating sequence resulted in a small overpotential of 0.26 V and a Tafel slope of 46 mV per decade [24]. Apart from graphene, other carbon materials [25,26], as well as copper and nickel meshes [4,5], are also optical candidates to composite with LDHs to improve their OER performance.

In this study, the composite of NiMn-LDHs and rGO loaded on NF was facilely prepared by a one-pot hydrothermal treatment of a mixed $Ni^{2+}$/$Mn^{2+}$ salt solution in the presence of urea and NF. When subjected to the test for OER, the as-prepared NiMn-LDHs/rGO/NF displayed an extremely low overpotential at 10 mA cm$^{-2}$ and a much smaller Tafel slope, which might be due to the synergistic effect established between NiMn-LDHs and rGO. rGO prevents the NiMn-LDH nanosheets from agglomeration, whereas the NiMn-LDH nanosheets alleviate the graphitization of rGO during hydrothermal process.

## 2. Results

### 2.1. Materials Characterizations

Shown in Figure 1 are the SEM images of NiMn-LDHs/NF and NiMn-LDHs/rGO/NF synthesized by hydrothermal method. It can be seen from Figure 1a that the synthesized NiMn-LDHs agglomerate on the surface of NF to form particles as large as 1 μm. In contrast, as displayed in Figure 1b, the ultrathin sheets of NiMn-LDHs are cross-linked and assembled vertically and uniformly on the surface of rGO to form nano-walled networks, which are certainly beneficial for the enlargement of the specific area of the synthesized

composite electrode. Therefore, rGO undoubtedly plays a role to prevent NiMn-LDH nanosheets from aggregating together into large particles.

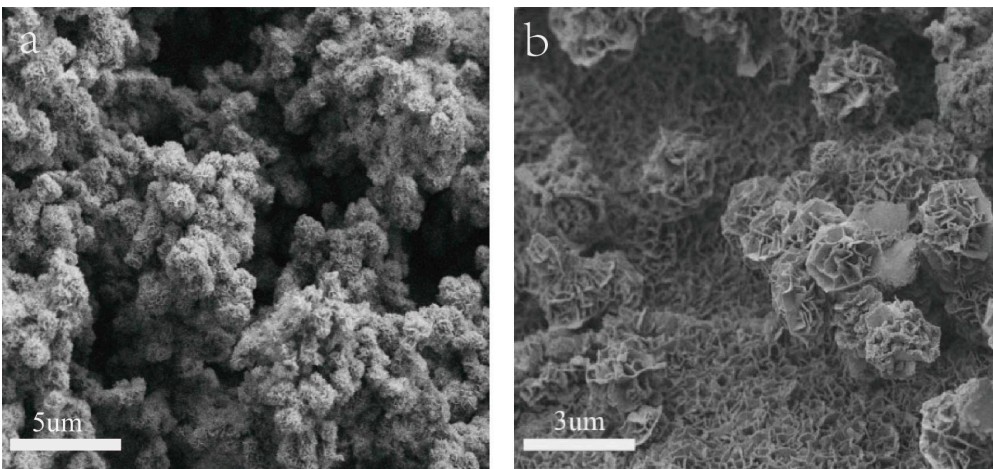

**Figure 1.** SEM images (**a**,**b**) of NiMn- LDHs/NF and NiMn-LDHs/rGO/NF.

TEM observations were applied to investigate the microstructure of the NiMn-LDHs/rGO composite deposited on the inner surface of NF. In the TEM image shown in Figure 2a, thin rGO sheets are observed with curved brims, whereas the ultrathin NiMn-LDH nanosheets are flat and transparent with straight and sharp edges. In the HR-TEM image displayed in Figure 2b, the lattice fringes of a NiMn-LDHs nanosheet are arranged regularly with a lattice spacing of 0.24 nm, which corresponds perfectly to the (110) crystallographic plane NiMn-LDHs [23]. Moreover, these highly oriented lattice fringes indicate that the synthesized NiMn-LDHs in the NiMn-LDHs/rGO composite are of a long-range-ordered crystal structure, which is strongly supported by the SAED image shown in Figure 2c. From the energy dispersive spectroscopy (EDS) elemental mapping image demonstrated in Figure 2d, it can be seen that the Mn, Ni, C and O elements are homogeneously distributed in the NiMn-LDHs/rGO composite, confirming the successful combination of rGO and NiMn-LDH sheets.

In the X-ray diffraction (XRD) pattern of NiMn-LDHs/rGO shown in Figure 3a, the indexed crystal planes of all diffraction peaks correspond to the typically layered structure of hydrotalcite-like materials (NiMn-LDHs: PDF 38-0715), and the broad peak centered around 23° is assigned to rGO. Thus, it is concluded that the NiMn-LDH sheets integrate with rGO with high purity and crystallinity, and that rGO has little effects on the crystalline structure of NiMn-LDHs. In the Fourier transform infrared (FT-IR) spectrum displayed in Figure 3b, the broad band centered around 3432 cm$^{-1}$ is ascribed to the superimposed stretching vibration of hydroxyl groups out of hydrogen-bonded $H_2O$ molecules intercalated between LDH layers and metal hydroxyl groups in NiMn-LDHs. The strong bands at 640 cm$^{-1}$ could be attributed to the vibrational stretching mode of M–O–M and M–O in hydrotalcite-like materials, and the intense band that appeared at 1380 cm$^{-1}$ is recognized to the asymmetric stretching mode of nitrate ions, intercalating in the interlayer [22,23]. On the other hand, the features located at 2229, 1657 and 1144 cm$^{-1}$ correspond to the C=O stretching, C=O ring stretching and C–O–C bending vibrations of rGO, respectively. In the Raman spectrum of NiMn-LDHs/rGO demonstrated in Figure 3c, the most prominent scattering bands are assigned to the D and G bands of rGO, and the features appeared at 515 and 596 cm$^{-1}$ are ascribed to the symmetric stretching vibrations of disordered (or defected) Ni(OH)$_2$ and MnOOH [27], respectively. In comparison with that of rGO, the intensity ratio of D band to G band (I$_D$/I$_G$) is obviously increased. Therefore, upon the formation of the NiMn-LDHs/rGO composite, the rGO is well exfoliated and enriched in structural defects, which contributes substantially to both the enlargement of a specific surface and to the increase in active sites. In summary, the data shown in Figure 3

strongly confirm that the NiMn-LDHs/rGO composite was successfully synthesized in the pores of NF [23].

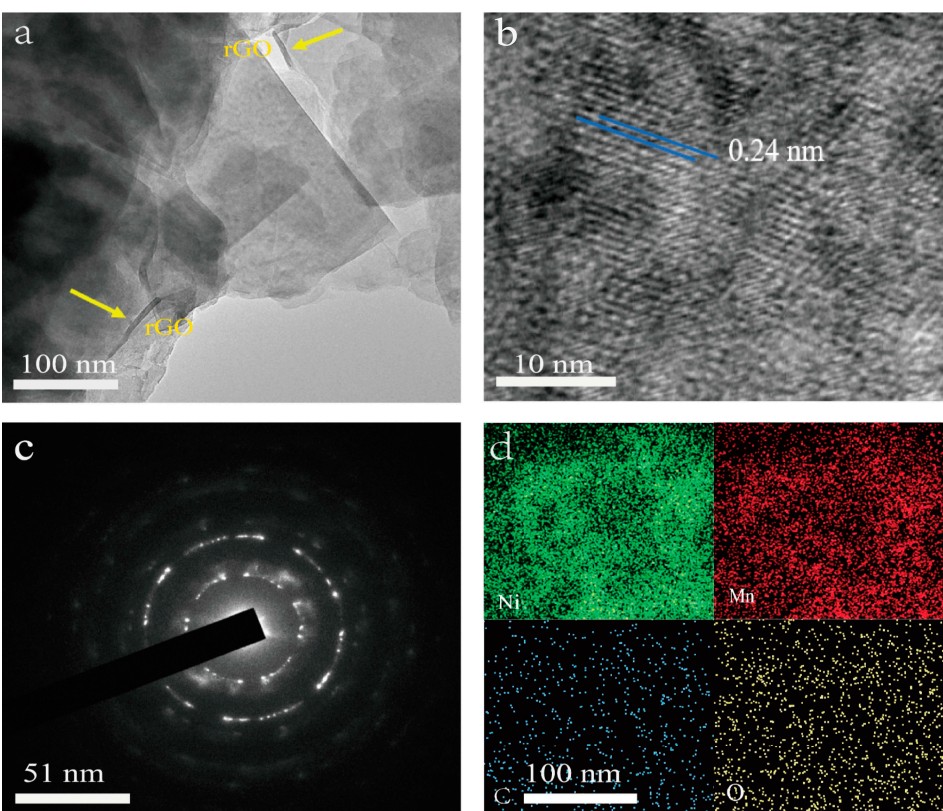

**Figure 2.** TEM, HR-TEM, SAED and TEM-EDS images (**a–d**) of NiMn-LDHs/rGO/NF.

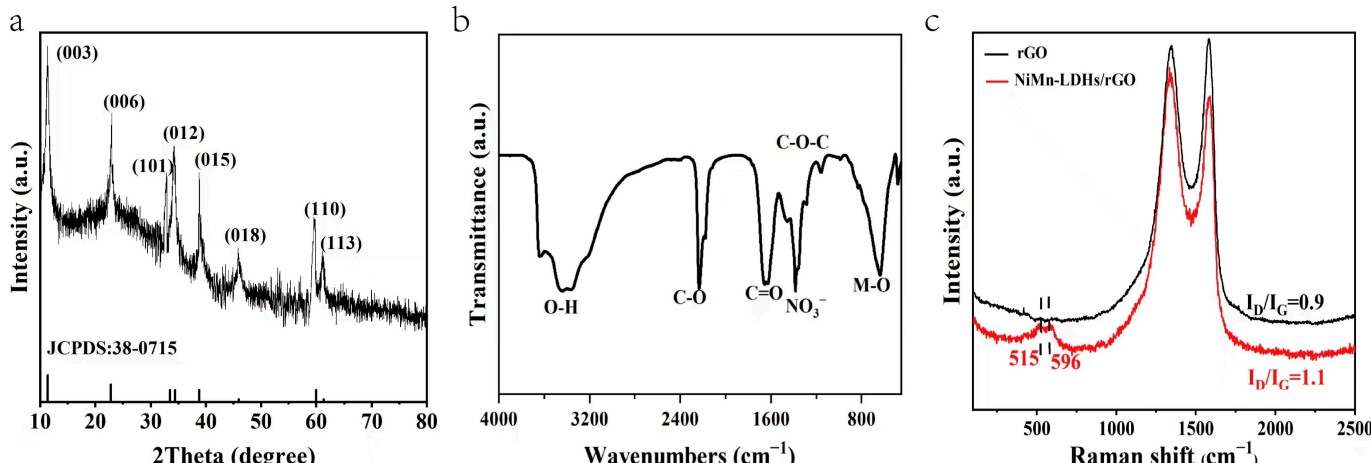

**Figure 3.** XRD pattern, FT-IR spectrum and Raman spectrum (**a–c**) of NiMn-LDHs/rGO.

The surface chemical state and elemental composition of the NiMn-LDHs/rGO composite was investigated by XPS. Illustrated in Figure 4a–d are the survey and deconvoluted Ni 2p, Mn 2p and C 1s XPS spectra of the NiMn-LDHs/rGO composite, respectively, along with the deconvoluted Ni 2p and Mn 2p ones of NiMn-LDHs for comparison. From the survey of the XPS spectrum displayed in Figure 4a, it can be seen that elements including Ni, Mn, C and O are present in the NiMn-LDHs/rGO composite, which is consistent with the results of TEM-EDS elemental mapping. In the cases of $Ni^{2+}/Ni^{3+}$ and $Mn^{2+}/Mn^{3+}$, it is difficult to assign a single binding energy (BE), even as a main peak center of gravity

(CG), to these chemical states because of the complex main line splitting due to multiplet contributions and satellite structures at higher BEs. Nonetheless, previous investigations support the assignment of the BEs of 854.6 and 856.1 eV in Ni $2p_{3/2}$ spectra to $Ni^{2+}$ and $Ni^{3+}$, respectively, in oxide, hydroxide and oxyhydroxide [28–30]. As displayed in Figure 4b, the characteristic peak of Ni $2p_{3/2}$ is located at 855.1 eV, and that of $2p_{1/2}$ is situated at 872.9 eV, while the respective shake-up satellites are located approximately 6 eV higher than the corresponding main peaks. In this text, we simply fit Ni $2p_{3/2}$ spectrum with three peaks instead of the Gupta and Sen (GS) $Ni^{2+}$ and $Ni^{3+}$ multiplet sets [31,32]. Aside from the satellite peak at 861.34 eV, the measured binding energies of Ni $2p_{3/2}$ can be divided into two peaks at 855.14 and 856.17 eV. The former is slightly larger than that of $Ni^{2+}$, whereas the latter is exactly the same as that of $Ni^{3+}$. Therefore, the $Ni^{2+}/Ni^{3+}$ ratio in the NiMn-LDHs/rGO composite is roughly calculated to be 1.26. Similarly, as demonstrated in Figure 4e, the characteristic peak of Ni $2p_{3/2}$ is located at 854.8 eV, and that of $2p_{1/2}$ is situated at 872.6 eV, and the measured binding energies of Ni $2p_{3/2}$ can be divided into two peaks at 854.60 and 855.76 eV. The former is exactly the same as that of $Ni^{2+}$, whereas the latter is slightly smaller than that of $Ni^{3+}$. Therefore, the $Ni^{2+}/Ni^{3+}$ ratio in the NiMn-LDHs is roughly calculated to be 2.07. The chemical state analysis of Mn cannot be performed simply based on its bonding energy in Mn 2p peak because the bonding energy difference for Mn with a different oxidation state is relatively small, and other features, including the shack-up satellite of Mn (II), the multiplet splitting and the peak shape, are necessary to be considered. As exhibited in Figure 4c (lower), the characteristic peak of Mn $2p_{3/2}$ is located at 643.2 eV, and that of $2p_{1/2}$ is situated at 653.8 eV. Of interest to note is that the peak shape of Mn $2p_{3/2}$ is similar to that reported for MnOOH, but quite different from those of MnO, $Mn_2O_3$ and $MnO_2$ [33], indicative of the presence of MnOOH. As displayed in Figure 4c (upper), the Mn $2p_{3/2}$ was fitted with the parameters from MnOOH [33], which was separated into eight components at 637.86, 639.54, 640.78, 641.89, 643.04, 644.28, 645.86 and 636.32 eV, respectively. Moreover, in comparison with those of NiMn-LDHs, the BEs of Ni 2p and Mn 2p of NiMn-LDHs/rGO are positively shifted to some extent. This is a result of the charge transfer from the metal ions in NiMn-LDHs to rGO and the increase in the number of high-valence $Ni^{3+}$ and $Mn^{3+}$ species, which afford more active sites for NiMn-LDHs/rGO to participate in the OER process. In the deconvoluted XPS C 1s spectrum (Figure 4d), the peaks located at 283.67, 284.29, 285.07, 287.50 and 288.30 eV could be attributed to the characteristic features of $sp^2$-hybridized graphitic carbon, $sp^3$-hybridized saturated carbon, graphitic carbon, C–OH, C–O–C and C=O bonds [34], respectively, implying that GO has been partly reduced to rGO. The residual oxygen atoms in rGO play a role of "oxygen bridge" for the charge transfer from the metal ions in NiMn-LDHs to rGO, i.e., the interactions between NiMn-LDHs and rGO contributes significantly to the enhancement of the OER catalytic activity of the NiMn-LDHs/rGO composite.

In their $N_2$ adsorption–desorption isotherms shown in Figure 5a, both NiMn-LDH and NiMn-LDH/rGO exhibit typical IV isotherms with a type H3 hysteresis loop, indicating that mesopores are randomly distributed and interconnected. Moreover, the calculated specific surface area of NiMn-LDH/rGO (43.18 $m^2$ $g^{-1}$) is much larger than that of NiMn-LDH (24.00 $m^2$ $g^{-1}$). Barrett–Joyner–Halenda (BJH) analyses shown in Figure 5b reveal the mesoporous structures of NiMn-LDH/rGO and NiMn-LDH, whose pore sizes are mainly distributed around 35.28 and 39.74 nm, with total pore volumes of 0.0034 and 0.0013 $cm^3$ $g^{-1}$, respectively. In comparison with that of NiMn-LDH, the average pore size of NiMn-LDH/rGO is decreased to some extent, which might be a result of the interactions between NiMn-LDH and rGO sheets.

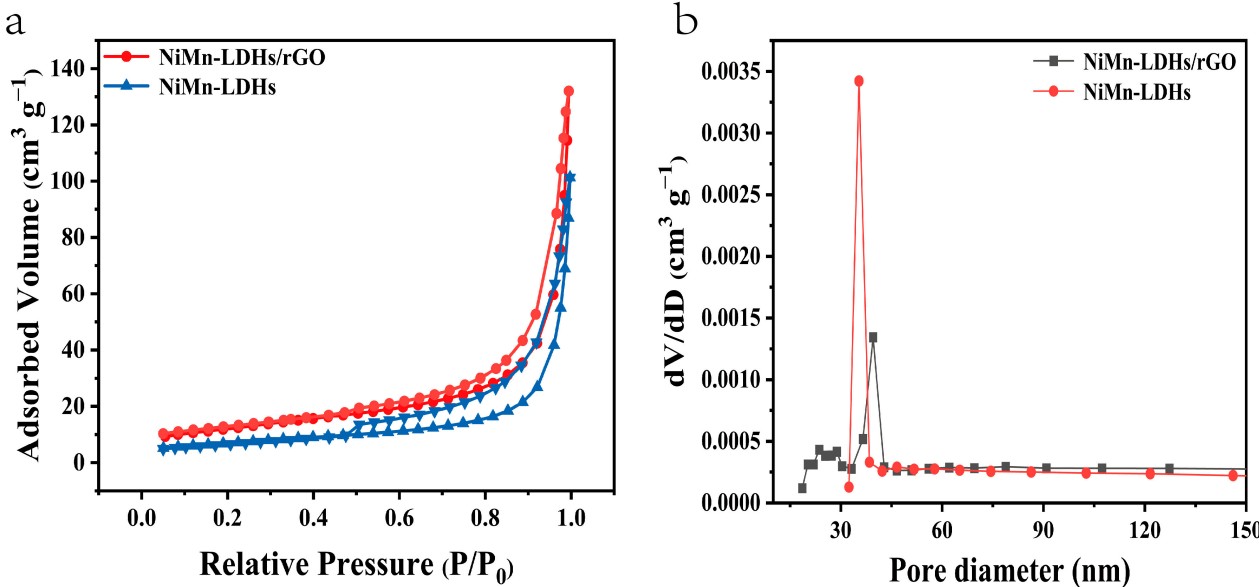

**Figure 4.** (**a**) The survey and (**b**–**d**) deconvoluted Ni 2p, Mn 2p and C 1s XPS spectra of the NiMn-LDHs/rGO composite along with (**e**,**f**) the deconvoluted Ni 2p and Mn 2p spectra of the NiMn-LDHs.

**Figure 5.** (**a**) $N_2$ adsorption–desorption isotherms and (**b**) BJH pore size distributions of NiMn-LDHs/rGO and NiMn-LDHs.

### 2.2. Electrocatalytic Performance

The optimization of the different amounts of composite components was conducted as follows. Firstly, the composite samples with different molar ratio (n) of Ni/Mn were prepared by hydrothermal treatment of 0.08, 0.12 or 0.16 mmol $Ni(NO_3)_2 \cdot 6H_2O$ with 0.04 mmol $MnCl_2$, 0.72 mmol urea and 0.6 mg rGO, respectively. The OER polarization curves of these samples were collected by linear sweep voltammetry. As shown in Figure 6a, the overpotentials for the $Ni_2Mn$-LDH/rGO/NF, $Ni_3Mn$-LDH/rGO/NF and $Ni_4Mn$-LDH/rGO/NF electrodes are 180, 140 and 200 mV, respectively, to achieve a current density of 10 mA cm$^{-2}$. Thus, the $Ni_3Mn$-LDH/rGO/NF sample demonstrated the best electrochemical performance. Secondly, the composite samples with different weight ratios (m) of rGO to $Ni_3Mn$-LDH were also prepared by hydrothermal treatment of 0, 0.6. 1.2 or 1.8 mg of rGO, with 0.12 mmol $Ni(NO_3)_2 \cdot 6H_2O$, 0.04 mmol $MnCl_2$ and 0.72 mmol urea, respectively. The OER polarization curves of these samples were collected by linear sweep voltammetry. As shown in Figure 6b, the overpotentials for the NiMn-LDH/NF, NiMn-LDH/rGO-0.04/NF, NiMn-LDH/rGO-0.08/NF and NiMn-LDH/rGO-0.16/NF electrodes are 220, 140, 170 and 210 mV, respectively, to achieve a current density of 10 mA cm$^{-2}$. Accordingly, the $Ni_3Mn$-LDH/rGO-0.04/NF sample, referred as NiMn-LDH/rGO/NF in the manuscript, was chosen as the research object, owing to its best electrochemical performance.

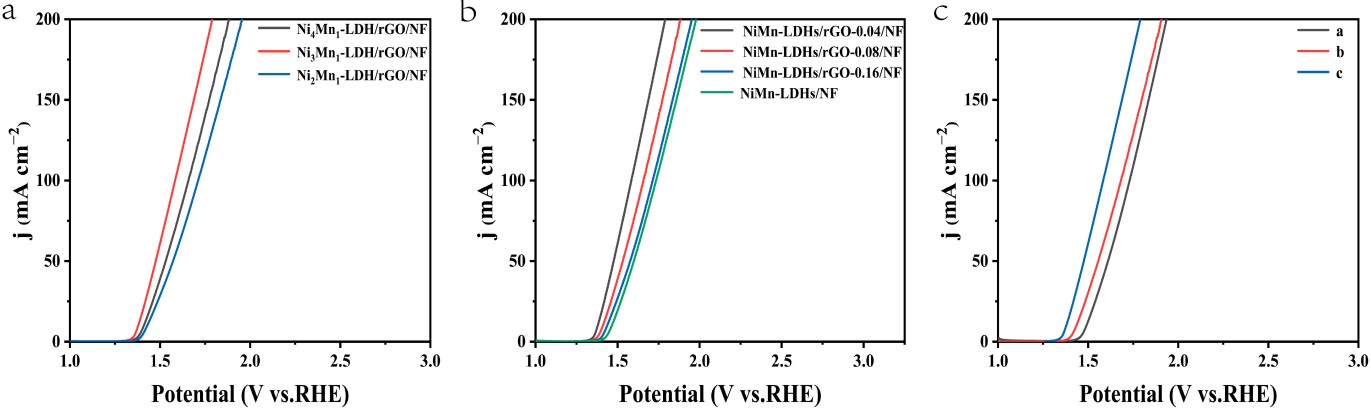

**Figure 6.** The OER polarization curves of (**a**) $Ni_nMn$-LDH/rGO/NF (n = 2, 3 and 4) and (**b**) NiMn-LDH/rGO-m/NF (m = 0, 0.04, 0.08 and 0.16) in 1 M KOH solution and those of (**c**) NiMn-LDHs/rGO/NF in 0.5 M $H_2SO_4$, 1 M PBS and 1M KOH, respectively.

From the LSV curves shown in Figure 6c, it can be determined that the overpotentials to achieve 10 mA cm$^{-2}$ of current density for the NiMn-LDHs/rGO/NF electrode are 250, 210 and 140 mV in 0.5 M $H_2SO_4$, 1 M phosphate-buffered saline (PBS) and 1M KOH solutions, respectively. It is evident that the overpotential to achieve 10 mA cm$^{-2}$ of current density decreases with the increase in the environmental pH value for the prepared electrode. Therefore, the OER performances of the prepared electrodes were extensively studied in an alkaline environment.

The electrochemical performance of the NiMn-LDHs/rGO/NF composite electrode, along with those of NiMn-LDHs/NF, $RuO_2$/NF and NF for OER, was evaluated in a typical three-electrode system using Hg/HgO and Pt wire as the reference and counter electrodes, respectively, in 1 M aqueous KOH solution. Figure 7a shows the iR-corrected linear sweep voltammetry (LSV) curves at a scan rate of 5 mV s$^{-1}$ in the potential range of 1.2−3.0 V (vs. RHE). It is notable that the NiMn-LDHs/rGO/NF electrode exhibits the highest polarization current, which is 1.37 V vs. RHE for achieving a current density of 10 mA cm$^{-2}$. Moreover, it can be seen from Figure 7b that an overpotential as low as 0.14 V is sufficient for the NiMn-LDHs/rGO/NF electrode to operate at 10 mA cm$^{-2}$ current density, which is lower than that of NiMn-LDHs/NF (0.22 V), $RuO_2$/NF (0.32 V), rGO/NF

(0.35 V) or NF (0.38 V) electrode, evidencing the relatively good catalytic efficiency of the NiMn-LDHs/rGO/NF electrode, owing to the synergistic effect induced by the direct and interfacial contact between ultrathin NiMn-LDH nanosheets and partly reduced rGO [24].

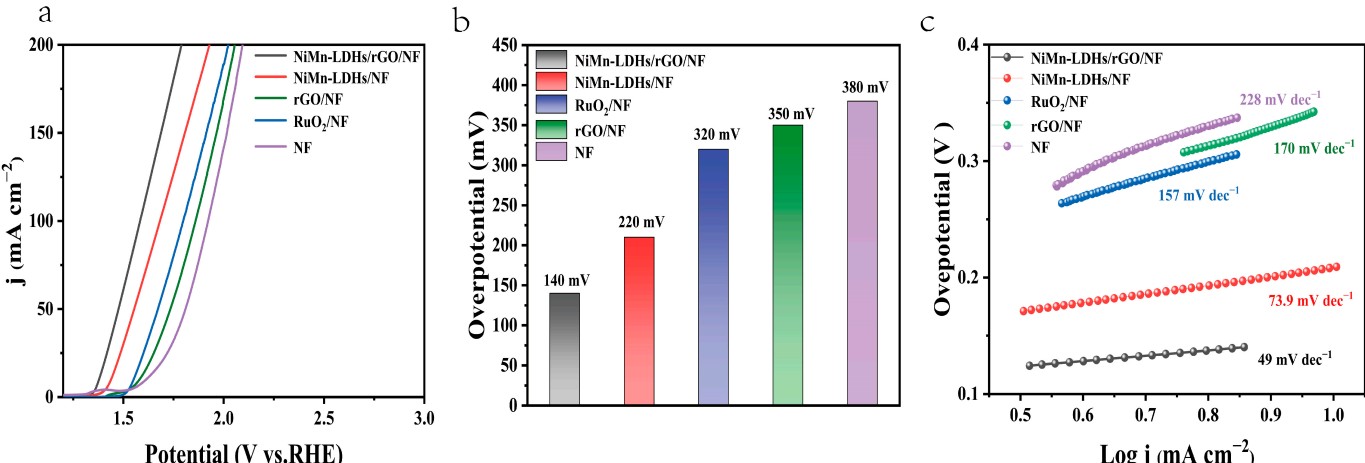

**Figure 7.** (**a**) NiMn-LDHs/rGO/NF, NiMn-LDHs/NF, RuO$_2$/NF, NF OER polarization curve. (**b**) Tafel diagram of above electrodes. (**c**) Overpotential diagram of above electrodes at 10 mA cm$^{-2}$.

The Tafel slope represents the reaction kinetics, which is another critical factor for evaluating the performance of a catalyst. As presented in Figure 7c, the Tafel slopes of the NiMn-LDHs/rGO/NF, NiMn-LDHs/NF, RuO$_2$/NF and NF electrodes are 49, 73.9, 157 and 228 mV dec$^{-1}$, respectively, implying the improved catalytic activity, i.e., the fast reaction kinetics, through the hetero-assembly of NiMn-LDH nanosheets and rGO. On the one hand, rGO can prevent NiMn-LDHs nanoflakes from agglomeration, offer more active sites and accelerate electron migration/transfer for the catalysis. On the other hand, the ultrathin NiMn-LDH sheets could also reduce the graphitization degree of rGO, leading to the increase in unsaturated sites and in the catalytic activity of the NiMn-LDHs/rGO composite catalyst. More importantly, the charge transfer from Mn and Ni ions of NiMn-LDHs to rGO leads to the increase in the valance state of Mn and Ni ions and the enlargement of charge distribution in rGO, rendering the NiMn-LDHs/rGO composite catalyst to be of the elevated conductivity, the synergistic coupling effect and, accordingly, the improved electrochemical performance. It can be seen from Table 1 that the NiMn-LDHs/rGO/NF composite electrocatalyst outperforms not only RuO$_2$/NF, but also most of the materials containing GO (rGO) or LDHs reported in the literature.

**Table 1.** The OER performances of materials containing GO (rGO) or LDH.

| Catalysts | Material Preparation [a] | Electrode Preparation | Overpotential (mV) [b] | Tafel Slope (mV/dec) | Cdl (mF/cm²) | Ref. |
|---|---|---|---|---|---|---|
| Co-NiMn LDH | HT | adhesive | 310 | 59 | -- | [13] |
| Co BTC-5wt%rGO | ST | adhesive | 290 | 60 | -- | [19] |
| N-NG-CoO | aerosolization | adhesive | 340 | 71 | -- | [20] |
| Cu MOF-GO | ST | adhesive | -- | 65 | -- | [21] |
| GO-FeNi LDH | ED | self-growth | 210 | 33 | 7.99 | [22] |
| GO/MnO$_2$-NiO | ED | self-growth | 379 | 47.84 | -- | [23] |
| NiMn-rGO | precipitation | adhesive | 260 | 46 | -- | [24] |
| Fe$_2$O$_3$@Ni-MOF | HT | deposition | 230 | 66.7 | 5.44 | [35] |
| Ni–Fe LDH | HT | adhesive | 219 | 39.5 | 0.334 | [36] |
| rGO/NF | HT | self-growth | 350 | 170 | 1.2 | This work |
| NiMn-LDH/rGO/NF | HT | self-growth | 140 | 49 | 3.38 | This work |

[a] HT = hydrothermal, ST = solvothermal, ED = electrodeposition; [b] at a current density of 10 mA cm$^{-2}$.

Moreover, the electrochemical stability is another criterion to evaluate the OER catalysts. A chronopotentiometry test under the current density of 10 mA cm$^{-2}$ was conducted to estimate the stability and durability of the composite electrocatalysts. As shown in Figure 8a, the current density of the NiMn-LDHs/rGO/NF electrode was kept nearly constant for 12 h. From Figure 8b, it can be seen that no obvious change is recorded for the original NiMn-LDHs/rGO/NF electrode and after 3000 CV cycles under the current density of 10 mA cm$^{-2}$, indicating that the NiMn-LDHs/rGO/NF composite catalyst has excellent stability and durability in alkaline solution. Such improvement may be a result of the high mechanical strength and electrochemical stability of the NiMn-LDHs/rGO/NF electrode.

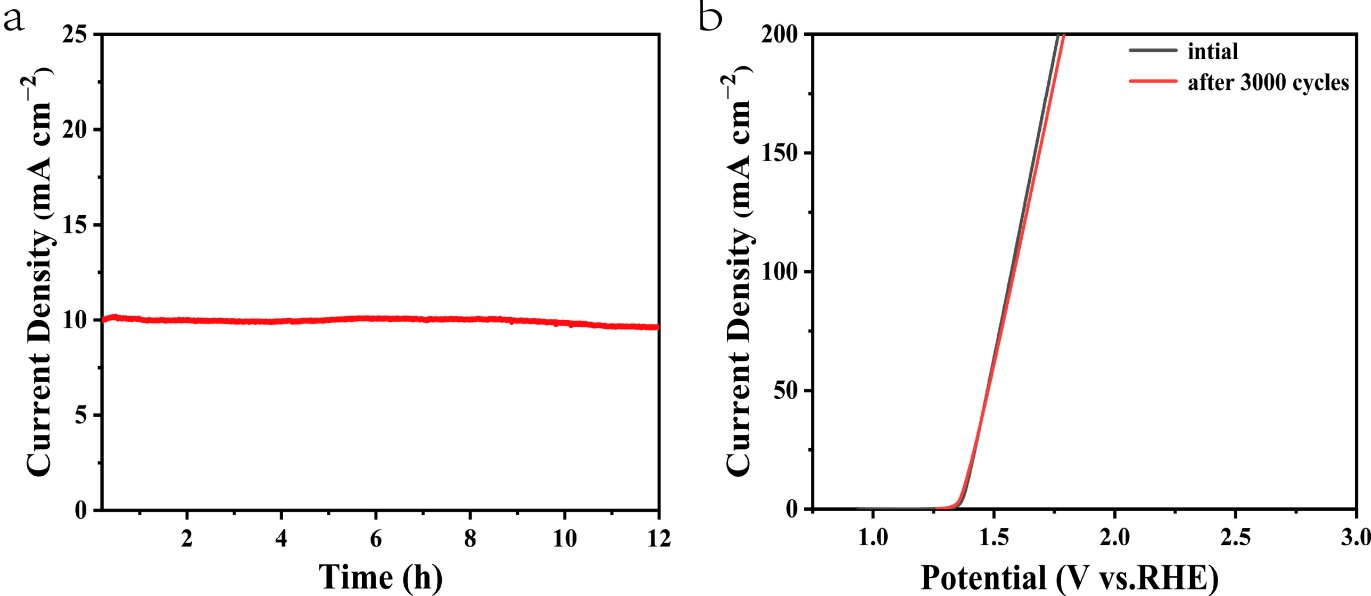

**Figure 8.** (**a**) Chronopotentiometry curve of the NiMn-LDHs/rGO/NF electrode at a constant current density of 10 mA cm$^{-2}$, (**b**) iR-corrected LSV curves of original catalysts (black line) and after 3000 CV cycles for overall water splitting in 1 M KOH under a current density of 10 mA cm$^{-2}$ (red line).

The electrochemical active surface area (ECSA) of the investigated catalysts were estimated via a simple CV scanning (see Figure 9a–d) for a better evaluation of the number of catalytic active sites. The slope of $\Delta j/2$ vs. scan rate is known to be equal to the value of Cdl (Cdl is the double-layer capacity), which has a linear relationship with ECSA [14,15]. From the fitted lines shown in Figure 9e, it is clear that the Cdl of the investigated catalysts increases gradually in the following order NF< rGO/NF (1.2 mF cm$^{-2}$) <RuO$_2$/NF (1.43 mF cm$^{-2}$) < NiMn-LDHs/NF (2.68 mF cm$^{-2}$) < NiMn-LDHs/rGO/NF (3.38 mF cm$^{-2}$). Because it has the highest value of ECSA, the NiMn-LDHs/rGO/NF composite catalyst should be one of the largest number of active sites, which is one of the crucial factors that determines its excellent OER performance. Since the Tafel slope obtained directly from the LSV curve is affected by the solution impedance and the material internal resistance, it reflects the overall OER kinetics of the catalyst, but cannot be used to analyze the OER mechanism of the catalyst. Thus, EIS was applied to obtain information on charge transfer between the electrolyte and the electrode surface to elucidate the kinetic differences of the catalysts investigated in this work. From the EIS spectra shown in Figure 9f, it can be seen that the NiMn-LDHs/rGO/NF composite electrode exhibits the smallest diameter among the Nyquist semicircles, which indicates that the electrode has the lowest charge transfer resistance (Rct). In contrast, NF displays the largest Rct, which is a result of a hydroxide/oxide layer generated spontaneously when it is exposed to air [37]. Since Rct influences the conductivity of the electrode and thus electron transfer in the electrochemical reaction process, it is expected that the NiMn-LDHs/rGO/NF

electrode also has the quickest charge-transfer rate during reaction process, and hence, the highest electrochemical activity among the investigated electrodes. Moreover, the catalysts which possess a layered atomic structure demonstrate a greatly decreased charge transfer resistance relative to $RuO_2/NF$, suggesting that the atomic structure greatly influences the electronic conductivity of the catalysts.

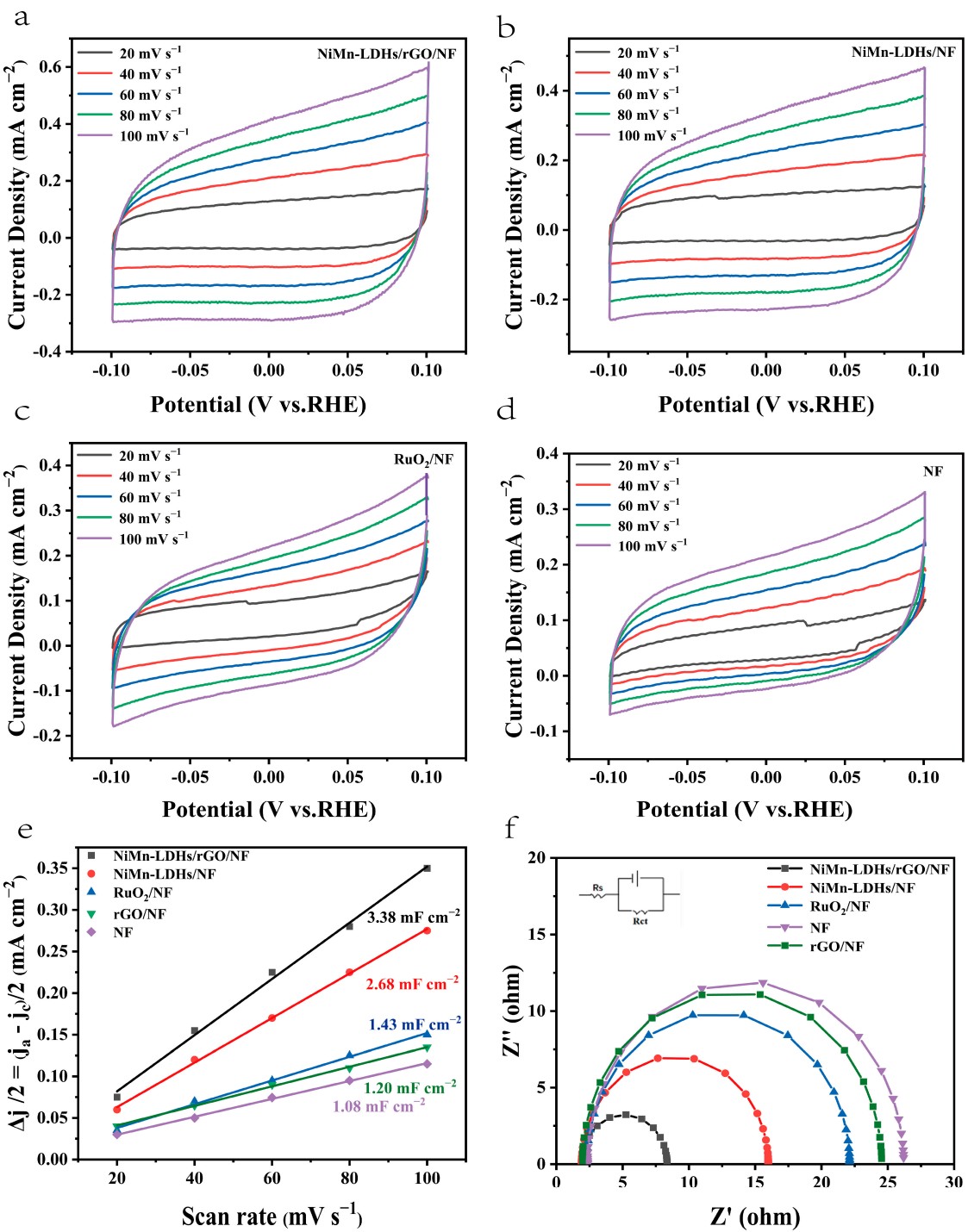

**Figure 9.** (**a**–**d**) CV curves in 1.0 M KOH, (**e**) linear plot of capacitance vs. scan rate and (**f**) Nyquist plots for NiMn-LDHs/rGO/NF, NiMn-LDHs/NF, $RuO_2/NF$, rGO/NF and NF.

In line with the adsorbate evolution mechanism (AEM) [38], the electro-catalyzing OER is a four-electron reaction process in alkaline electrolyte, which involves multiple

reaction intermediates ($M^- * OH$, $M^- * O$, $M^- * OOH$ and $O_2$, $M^- *$ represents the active site of the reaction) generated via the multiple reaction steps shown below:

$$M^- * + OH^- \rightarrow M^- * OH + e^-$$
$$M^- * OH + OH - \rightarrow M^- * O + H_2O + e^-$$
$$M^- * O + OH^- \rightarrow M^- * OOH + e^-$$
$$M^- * OOH + OH^- \rightarrow M^- * + O_2 + H_2O + e^-$$

(1)

In each OER reaction step, a large overpotential is required for its high reaction energy barrier, which leads to slow reaction kinetics. LDHs have broad potentials in OER because of their unique 2D-layered structures, good stability and low cost. However, their poor conductivity and limited reactive sites prevent them from industrial application. Aiming at improving the catalytic behavior and accelerating the catalytic reaction kinetics of LDHs, the self-supporting NiMn-LDH/rGO/NF composite electrode was constructed through the growth of NiMn-LDH and rGO sheets on NF.

The TEM images shown in Figure 2 confirm that the NiMn-LDHs/rGO composite is composed of ultrathin NiMn-LDHs and thin rGO nanosheets. The intercalation of rGO restricts the self-aggregation of NiFe-LDH sheets and vice versa. With the decrease in the thickness of nanosheets, the abundant heterogeneous interfaces between NiMn-LDHs and rGO cause more catalytic active sites to be exposed to the electrolyte solution. Meanwhile, the structural defects in ultrathin NiMn-LDHs and thin rGO nanosheets offer additional active sites to the promotion of OER reaction. Therefore, the high density of catalytic active sites in the NiMn-LDHs/rGO/NF composite are responsible for its accelerated OER surface reaction kinetics and its greatly improved electro-catalyzing OER performance. Relative to those of NiMn-LDHs, the much larger specific surface area (see Figure 5) and electrochemical active surface area (see Figure 9e) of the NiMn-LDHs/rGO composite denote a solid support to this viewpoint. On the other hand, the 2D rGO, 3D arraylike structure of NiMn-LDHs and 3D NF constitute a conductive framework, which is beneficial to promote the penetration of electrolyte ions, reduce charge transfer resistance (see Figure 9f) and generate a fast transfer path for electrons/ions in the reaction process. Thus, the conductive framework in the NiMn-LDH/rGO/NF composite electrode facilitates both the increase in conductivity and the contact between active site and electrolyte, resulting in the acceleration of the OER reaction kinetics. Last but not least, the two-dimensional NiMn-LDH and rGO form a van der Waals heterojunction due to their interfacial electrostatic adsorption [39], hydrogen bonding (H-bond) effect [40–42] and terminal oxo-complexation [43,44]. Since the Fermi energy level of NiMn-LDH (−4.2 eV for FeNi-LDH) [45] is higher than that of rGO (−5.1 eV for GO) [46], once contact is made with each other, electrons will transfer from NiMn-LDH to rGO to reach a thermal equilibrium state. Consequently, the positively charged n-type NiMn-LDH is conducive to reduce the adsorption energy of hydroxide anions and reaction intermediates on active sites and to promote the electron transfer from oxygen to metal atoms, which facilitates the electro-catalyzing of OER on the self-supporting NiMn-LDH/rGO/NF composite electrode.

In line with the discussion on the OER mechanism, the synergistic effect of catalyst components was summarized. As displayed in Figure 7, the electro-catalyzing OER performance continuously improves for the NF, rGO/NF and NiMn-LDHs/rGO/NF electrodes, which could be ascribed to the synergistic effect of catalyst components, i.e., the synergistic coupling of NiMn-LDH, rGO and NF. As explained above, the successful integration of ultrathin NiMn-LDH, partly reduced rGO and 3D-framed NF leads to a large increase in the number of active sites, an efficient charge/mass transfer and the easy adsorption of hydroxide anions and reaction intermediates on active sites, which are responsible for the substantially enhanced electrocatalytic activity of the composite catalyst.

## 3. Materials and Methods

### 3.1. Materials

$Ni(NO_3)_2 \cdot 6H_2O$, $MnCl_2$, $RuO_2$, nickel foam, acetone and hydrochloric acid were obtained from Aladdin Bio-Chem Technology Co. Ltd. (Shanghai, China), Potassium permanganate was acquired from Liaoning Quanrui Reagent Co (Jinzhou, China), and urea from from Beijing Chemical Works TJCCR Co. Ltd. (Beijing, China). All chemicals used in this work were of analytical grade and used without further purification.

### 3.2. Preparation of Graphene Oxide (GO)

Graphene oxide (GO) was prepared from Specpure graphite powder by a modified Hummers' method [47,48]. In brief, 1 g of graphite powder was dispersed into 23 mL $H_2SO_4$ (95%) in an ice bath, and 3 g $KMnO_4$ was slowly added under vigorous stirring. After the mixture was ultrasonicated at room temperature for 10 h, 46 mL deionized water was carefully injected to obtain a brown solution, then 30% $H_2O_2$ was dropped in until the mixture turned to bright yellow. The reactant mixture was filtered and then washed with 0.2 M HCl until the pH value of the filtrate was around 6. The exfoliated GO was achieved when the filter cake was dried in vacuum at 323 K for 24 h.

### 3.3. Preparation of the Self-Supporting NiMn-LDHs/rGO/NF Composite Electrode

The procedure for the preparation of the self-supporting NiMn-LDHs/rGO/NF composite electrode is depicted in Figure 10. GO was prepared by oxidation of graphite powder using Hummers' method. A mixture containing 0.12 mmol $Ni(NO_3)_2 \cdot 6H_2O$, 0.04 mmol $MnCl_2$ and 0.72 mmol urea was dissolved in 20 mL of deionized water, and then 10 mL of 0.2 mg/mL rGO solution was added. After ultrasonication for 40 min, the solution along with a piece of NF (10 × 10 mm)—which was ultrasonically cleaned in 2 M HCl, acetone, ethanol and deionized water, respectively, and then dried at 333K for 5 h—was poured into a 50 mL Teflon lining autoclave and kept at 140° for 10 h. When the autoclave was cooled naturally to room temperature, the product was collected and washed with a large amount of deionized water to remove the adsorbed substances. The self-supporting NiMn-LDHs/rGO/NF composite electrode was achieved after drying the product in an oven at 60° for 3 h. The self-supporting NiMn-LDHs/NF composite electrode was prepared in the same procedure without the addition of rGO. For comparison, the $RuO_2$/NF electrode was also prepared following the method previously reported [49].

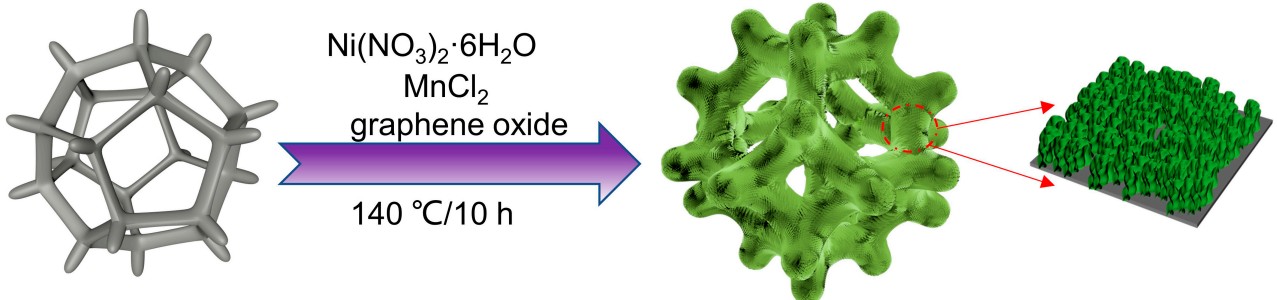

$Ni(NO_3)_2 \cdot 6H_2O$
$MnCl_2$
graphene oxide

140 °C/10 h

**Figure 10.** Schematic description of the procedure for the preparation of NiMn-LDHs/rGO/NF composite catalyst.

### 3.4. Characterization

The morphology of the catalysts was observed by scanning electron microscopy (SEM, QUANTA 200S, FEI, Eindhoven, the Netherlands) and transmission electron microscopy (TEM, JEM2100, JEOL, Tokyo, Japan). The crystalline phase of the catalyst was obtained by X-ray diffraction (XRD, D8 Advance, Bruker, Berlin, Germany) at 40 kV and 200 mA operating conditions in the 2θ range from 10 to 70. The elemental composition and oxidation state of the catalysts were analyzed by X-ray electron spectroscopy (XPS, KRATOS, Stratford,

UK). Raman scattering spectra were collected by a Jobin Yvon (Palaiseau, France) HR 800 micro-Raman spectrometer with 532 nm excitation. Fourier transform infrared spectra (FT-IR) were recorded by a Perkin Elmer (Waltham, MA, USA) Spectrum 100FT-IR spectrometer with background correction by referring KBr pellets.

All of the electrochemical tests were conducted on an SP-300 electrochemistry work-station (Bio-Logic, Seyssinet-Pariset, France) in a three-electrode system. The reference electrode was an Hg/HgO one, and the counter electrode was a sheet of platinum. The investigated electrodes, including NiMn-LDHs/rGO/NF, NiMn-LDHs/NF, RuO$_2$/NF and NF, were used as working electrodes, respectively. The electrolyte was a solution of 1 mol L$^{-1}$ KOH, which was bubbled with pure oxygen for 30 min to maintain oxygen saturation. The potentials achieved were converted into reversible hydrogen ones by the following equation:

$$E_{RHE} = E_{Hg/HgO} + 0.059 \times pH + 0.098 \tag{2}$$

Prior to data collection, cyclic voltammetry (CV) scanning was performed for 30 min at a scanning rate of 100 mV s$^{-1}$ to fully activate the working electrode. CV measurements were conducted in the range of $-0.1$ V to 0.1 V (vs. Hg/HgO electrode). Linear scan voltammetry (LSV) polarization curves in the voltage range of 0 to 1 V for OER reactions were obtained at a scanning rate of 5 mV s$^{-1}$ in a solution of 1 mol L$^{-1}$ KOH. Electrochemical impedance spectroscopy (EIS) was performed over a frequency range spanning from 100 kHz to 0.01 Hz. The transient photocurrent–time curves (I–t) were applied to assess the stability of the prepared materials. The value of Cdl (Cdl is the double-layer capacity) equals to the slope of the fitted straight line in the plot of the current density difference vs. scanning rate, which has a linear relationship with the interfacial area between the electrode surface and the electrolyte. Thus, ECSA can be extracted from the electrochemical double-layer capacitance (Cdl) by:

$$ECSA = Cdl/Cs \tag{3}$$

where, Cs represents the specific capacitance, which is 0.040 mF cm$^{-2}$ in a solution of 1.0 M KOH, as previously reported [1].

## 4. Conclusions

An effective OER electrocatalyst was produced by anchoring the composite of NiMn-LDHs ultrathin nanosheets and thin-layered reduced graphene oxide on the skeleton of nickel foam via a facile one-pot hydrothermal method. A series of electro-chemistry experiments confirmed the excellent OER catalytic performance of the self-supporting NiMn-LDHs/rGO/NF composite electrode. In comparison with those of the self-supporting NiMn-LDHs/NF composite electrode, the overpotential at 10 mA cm$^{-2}$ and Tafel slope of the self-supporting NiMn-LDHs/rGO/NF composite electrode are deduced to be about 80 mV and 25 mV dec$^{-1}$, respectively, in a solution of 1 mol L$^{-1}$ KOH. The interactions between NiMn-LDHs and rGO are responsible for the largely enhanced electrocatalytic performance. On the one hand, ultrathin NiMn-LDH sheets reduce the graphitization degree of rGO, whereas thin-layered rGO also effectively prevents the re-aggregation of ultrathin NiMn-LDH sheets. On the other hand, a quick charge transfer from ultrathin NiMn-LDH sheets to thin-layered rGO occurs at their interfaces, causing both to rise for the binding energy and for the oxidation state of Ni and Mn atoms. Thus, it is expected that the conductivity and chemically active sites are accordingly increased, which is favorable to the enhancement of the electrocatalytic performance of the self-supporting NiMn-LDHs/rGO/NF composite electrode in the OER reaction.

**Author Contributions:** Conceptualization, J.W. and Y.L.; methodology, J.W.; validation, J.W.; formal analysis, J.W. and Y.L.; investigation, J.W.; resources, Y.L.; data curation, J.W. and Y.L.; writing—original draft preparation, J.W.; writing—review and editing, J.W. and Y.L.; supervision, Y.L. All authors have read and agreed to the published version of the manuscript.

**Funding:** This research received no external funding.

**Data Availability Statement:** Not applicable.

**Conflicts of Interest:** The authors declare no conflict of interest.

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
