# Peer review of "The Self-Supporting NiMn-LDHs/rGO/NF Composite Electrode Showing Much Enhanced Electrocatalytic Performance for Oxygen Evolution Reaction"

_catalysts, doi:10.3390/catal13061012_

Round 1

Reviewer 1 Report

It is a relatively good article and the article can be accepted with some suggestions.
1- The most important problem of the article is the number of catalyst samples. Only one sample has been synthesized! Authors should optimize different amounts of composite components. (Ni-Mn- rGO- NF)
2- OER mechanism is very weak. A separate part should be discussed about the mechanism.
3- Research results should be compared with similar articles in the table.
4- Add the BET analysis to the article.
5- An in-depth discussion is necessary about the synergistic effect of catalyst components.
6- It is better to repeat the tests in sulfuric acid and a neutral environment. (Report at least one LSV from these environments)
7- Describe Hummers' method in the text of the article by providing references.

 Minor editing of English language required

Reviewer 2 Report

This paper by Wang et al. prepared the NiMn-LDH/rGO on nickel foam and used as an oxygen evolution electrode material. This electrode material synthesis, characterization, and applications are not new. It seems like regular OER work other than the originality of the data. But, the authors explained systematically their catalyst preparation and OER applications. However, some points are missing in the manuscript. Therefore, before considering the publication of this work, the following important points should be solved.

1.      Figure 4. XPS deconvolutions are not followed the fitting rule. All the regions Ni 2p, Mn 2p, and C 1s need to be refit with controlled FWHM, otherwise fitting is meaningless.

2.      Page 6. “The residual oxygen atoms in rGO play an important role for the charge transfer from the metal ions in NiMn-LDHs to rGO, i.e., the interactions between NiMn-LDHs and rGO contributes a lot to the enhancement of the OER catalytic activity of the NiMn-LDHs/rGO composite. Moreover, in comparison with the features of NiMn-LDHs displayed in Figure 4e and Figure 4f, the binding energy of Ni2+ and Mn3+ of NiMn-LDHs/rGO is positively shifted about 0.4 eV, confirming the charge transfer from the metal ions of NiMn-LDHs to rGO.” This is author hypothesis, it should be explained with proper reference.

3.      To emphasize the present work, the author should compare earlier reports on LDH/rGO composite electrode materials synthesis method, ECSA, overvoltage for OER, and Tafel slope values.

4.      To authenticate the NiMn-LDHs/rGO OER performance. Compare the rGO/NF OER and electrochemical active surface area.   

5.      The EIS of NF shows high charge transfer resistance, it is a strange result. Bare NF is a metallic conductor. But the author claimed high charge transfer resistance, it should be explained.   

Minor comments

1.    XRD planes should be in parentheses

2.    The tick labels and axis labels are very hard to visualize, the author should improve the visibility of image quality and increase the axis font size.

Round 2

Reviewer 1 Report

The authors have addressed all concerns well. The article is acceptable in its current form.

Minor editing of English language required

Reviewer 2 Report

The authors fairly addressed all the comments; it can be acceptable in the “catalysts.”